

# Technical note: A time-integrated sediment trap to sample diatoms for hydrological tracing

Jasper Foets[1,2], Carlos E. Wetzel[1], Núria Martinez–Carreras[1], Adriaan J. Teuling[2], Jean-François Iffly[1], and Laurent Pfister[1,3]

[1]Environmental Research and Innovation Department, Luxembourg Institute of Science and Technology, Belvaux, Luxembourg
[2]Department of Environmental Sciences, Wageningen University and Research, Wageningen, Netherlands
[3]Faculty of Science, University of Luxembourg, Technology and Medicine, Belval, Luxembourg

**Correspondence:** Jasper Foets (jasper.foets@list.lu)

**Abstract.** Diatoms, microscopic, single-celled algae, are present in almost all habitats containing water (e.g. streams, lakes, soil, rocks) and form one of the most common and diverse algal groups in both freshwaters and marine ecosystems. In the terrestrial environment, their diversified species distributions are mainly controlled by physiographical factors and anthropic disturbances. This makes them useful tracers in catchment hydrology. In their use as a hydrological tracer, diatoms are generally

sampled in streams by means of an automated sampling method and as a result many samples are collected to cover a whole storm run-off event. As diatom analysis is labour intensive, a trade-off has to be made between the number of sites and the amount of samples per site. A potential way to reduce this number is by using a time-integrated mass-flux sampler. Here, we explored the potential for the Phillips sampler to provide a representative sample of the diatom assemblage of a whole storm run-off event. We addressed this by comparing the diatom community composition of the Phillips sampler to the composite

community collected by the automatic samplers for three events. Our results indicate that during two events the Phillips sampler sampled representative samples, whereas significantly different communities were collected during the third event. However, sediment data of this event, which was sampled with automatic samplers, showed much noise meaning that we could not verify if the Phillips sampler sampled representative communities or not. Nevertheless, we believe that this sampler could not only be applied in hydrological tracing using terrestrial diatoms, but may also be a useful tool in water quality assessment.

# 1 Introduction

Tracing of water sources and flow paths is an important field of study in catchment hydrology. Environmental tracers such as geochemical elements and stable isotopes of hydrogen and oxygen in the water molecule have led to new insights in our understanding of the age, origin and pathways of water at the watershed scale during the last few decades. For instance, the work of Hooper et al. (1990) on end-member mixing analysis showed that stream water could be distributed into unique water

signatures that mapped back to measurable water chemical concentrations in the landscape. Likewise, thanks to the use of hydrological tracers, we know that groundwater fulfils a very important role in the storm- and snow-melt run-off generation in streams (Sklash and Farvolden, 1979). However, progress remains stymied by various assumptions and limitations in the tech-


niques, including temporally varying input concentrations (McDonnell et al., 1990), unstable end-member solutions (Elsenbeer et al., 1995; Burns, 2002) and the need for unrealistic mixing assumptions (Fenicia et al., 2010). In response to the need for

new tracers, soil diatoms (i.e. diatoms living on the soil surface) have been proposed by Pfister et al. (2009) as a potential tracer for studying hydrological connectivity and surface run-off processes.

Diatoms are microscopic, eukaryotic, unicellular algae and form one of the most common and diverse algal groups in both freshwaters and marine environments (Round et al., 1990). They are pigmented, photosynthetic and because of their high abundances, they play a large role in the exchange of gasses between the atmosphere and biosphere. It has been estimated that

they are responsible for 20 % of the total oxygen production on the planet (Scarsini et al., 2019). The characteristic feature of diatoms is their siliceous, highly differentiated cell wall which shows an enormous diversity in shapes and structures. These species-specific cell wall ornamentations enable the identification of diatom taxa and form the basis of diatom taxonomy and systematics. Furthermore, their small cell sizes, commonly varying between 10 and 200 $\mu$m in diameter or length, allows them to be easily transported by flowing water within or between elements of the hydrological cycle (Pfister et al., 2009). Besides

their high abundances in aquatic environments, they are also present in most terrestrial habitats where their diversified species distributions are mainly controlled by physiographical factors (e.g. pH, moisture... see Lund, 1945; Van de Vijver et al., 2002) and anthropic disturbances (Antonelli et al., 2017; Foets et al., 2020). In a recent study Foets et al. (2020), we showed that indicator species could be assigned to certain land use types (e.g. agricultural field, forest...), whereas in another study we indicated that the absolute abundances of soil diatom were related to available soil moisture content (Foets et al., in review).

All these aforementioned characteristics make them a useful environmental marker and tracer in catchment hydrology.

In their use as a hydrological tracer, drift diatoms from both aquatic and terrestrial origin are generally sampled by means of an automated sampling method. This allowed researchers to follow the species abundances along the hydrograph and link these changes with the activation of hydrological connectivity. An interesting outcome was that the percentage of terrestrial diatoms in the stream samples increased when a peak in the hydrograph occurred meaning that they are indeed responsive to

changes in stream flow conditions (Klaus et al., 2015; Martínez-Carreras et al., 2015; Pfister et al., 2015). As a follow-up to this proof-of-concept work, Pfister et al. (2017) have advocated to better characterize the spatial and temporal dynamics of the terrestrial diatom reservoir - paving the way for new diatom sampling protocols of higher spatial and temporal resolution (Klaus et al., 2015). This should eventually provide new insights into the initiation of hydrological connectivity across the catchment (as inferred from diatom flushing to the stream from various terrestrial reservoirs). However, this is not possible at the moment

since diatom analysis is labour intensive (sample processing, cell identification and counting) and new, faster identification techniques (e.g. molecular tools) are not yet "standardized" or widely used (Keck et al., 2018; Rivera et al., 2020). So, when using automatic samplers a trade-off has to be made between the number of sites and the amount of samples per site.

A way to increase the number of sampling sites could be by using time-integrated passive samplers. As the name already suggests, they are designed to collect samples that are representative for the whole sampling period. An example is the time-

integrated mass-flux sampler designed by Phillips and co-workers (Phillips et al. (2000); Phillips sampler). This sampler, consisting of a PVC-pipe (98 mm x 1 m) with a very small in- and outlet (4 mm), has originally been developed to trap fine-grained (< 62.5 $\mu$m) suspended sediment for the assessment of geochemical, physical and magnetic features of transported



material (Russell et al., 2000). The functionality of the sampler is based on the large difference in diameter between the inlet of the pipe and the pipe itself. This sudden change in pipe diameter substantially reduces flow velocity and encourages

sedimentation of fine particles without significantly disrupting stream flow. This sampler is seen as a simple, inexpensive and easy deployable method particularly for studies requiring a dense sampling network (Phillips et al., 2000). Generally, studies illustrated that the Phillips sampler gives a representative sample of the sediment properties (Russell et al., 2000; Laubel et al., 2002; Walling, 2005) and could therefore for example be used for sediment source fingerprinting (Martínez-Carreras et al., 2010, 2012). However, its use is rather restricted to small streams (Phillips et al., 2000). Although the design could be easily

adapted (e.g. increase length and diameter of the pipe) depending on the river flow characteristics, the outcome has not always been positive (McDonald et al., 2010). Furthermore, it is not suited for assessing absolute sediment bed loads and since the sampler grossly underestimate the total fine sediment flux, the samples are also biased towards larger particles (Laubel et al., 2002; Perks et al., 2014). Besides its limitations, this low-cost, easy deployable sampler has reportedly provided satisfying results and as diatoms are a part of suspended sediment (Droppo, 2001) and have mean cell sizes comparable to silt particles

(Jones et al., 2012), one would expect that they also behave similarly in stream bodies. Therefore, the sampler could potentially also collect a diatom community representative of the sampling period.

However, diatoms also differ in several aspects of their behaviour from suspended silt particles that could affect the sampling efficiency of the Phillips sampler. First, they have a smaller mass density than silt particles, because their cells do not only consist of silica (generally ranging between 10 and 72 % depending on the environment and species (Round et al. (1990)),

but also contain organic material such as protoplasm and polysaccharides. Secondly, their mass density may further decrease due to an increase in thickness of a low-density mucilage sheath or envelope around the cell or colonial unit (Hutchinson, 1967). Additionally, the cell shape and elaborations to the shape will influence their sinking rate. For instance, a unit (colony or individual cell) with a high surface to volume ratio will have a higher viscous drag on the cell as it sinks and hence slow down its descent (Round et al., 1990). On the other hand, moving on or being attached to sediment particles, diatoms produce an

extracellular matrix of mucopolysaccharides, which will bind the particles and could eventually form aggregates (Gerbersdorf et al., 2008). Because of that, these attached cells will behave like a larger particle instead. Finally, phytobenthic diatoms in running waters exhibit a circadian diurnal activity rhythm in which they detach, drift and reattach to a substrate some centimetres or decimetres downstream (Müller–Haeckel, 1973; Müller-Haeckel and Håkansson, 1978). Overall, these different features are often species-specific, meaning that some species could be oversampled while others would be underestimated.

Here, we explore the potential for the Phillips sampler to provide a representative sample of the diatom assemblage of a whole storm run-off event. We addressed this by comparing the diatom community composition of the Phillips sampler to the composite community collected by the automatic samplers. As a control, we also checked if the particle size distribution of the suspended sediment collected by the Phillips sampler was similar to the composite distribution collected by the automatic sampler. We consider automatic samplers as sampling fully representative samples of a storm run-off event.





## 2 Methodology

### 2.1 Study site

The sampling location was situated in the south of the Colpach catchment (19.44 km$^2$; 49°46'27" N, 5°48'56" E), a sub-catchment of the Attert experimental basin (249 km$^2$) and located in the western part of the Grand Duchy of Luxembourg. The sub-catchment is mainly underlain by Devonian slate, phyllites and quartzite in the north and marls in the southern part (Fig. 1). Land-use in the area consists of forests (50 %), grass- (24 %) and cropland (23 %) (Juilleret et al., 2011). The altitude in the sub-cathment ranges between 265 m and 530 m, with our sampling site at 313 m a.s.l. Hydrology in a headwater creek of the Colpach catchment, with similar bedrock geology, topography and land use, has been characterised by a fill- and spill system (Wrede et al., 2015) with rapid flow in a highly permeable saprolite layer of weathered schist above the bedrock as the dominant run-off process. Besides the importance of lateral flow, high surface infiltration rates have been observed in the sub-catchment (Van den Bos et al., 2006). For a more detailed description on the hydrological features of the Colpach please see Loritz et al. (2017).

We chose a small, shallow part of the Colpach stream at the outlet of the catchment to conduct our experiment. The stream has a width of around 3 m and had a mean discharge of 0.27 m$^3$ s$^{-1}$ during 2018. Previously, Martínez-Carreras et al. (2010, 2012) successfully deployed a Phillips sampler at this location. In total, three storm run-off events were sampled between October and December 2018. The first event occurred between 30 October and 5 November, the second between 9 and 17 November and the third between 7 and 12 December. The summer period prior to sampling was exceptionally dry and warm with an average monthly precipitation of 60.8 mm (Meteolux, 2019), whereas the precipitation during October, November and December was 164.8, 138.8 and 144.9 mm respectively (data retrieved from weather station in Useldange, Administration des Services Techniques de l'agriculture (ASTA)). The Colpach has a temperate, semi-oceanic climate regime.

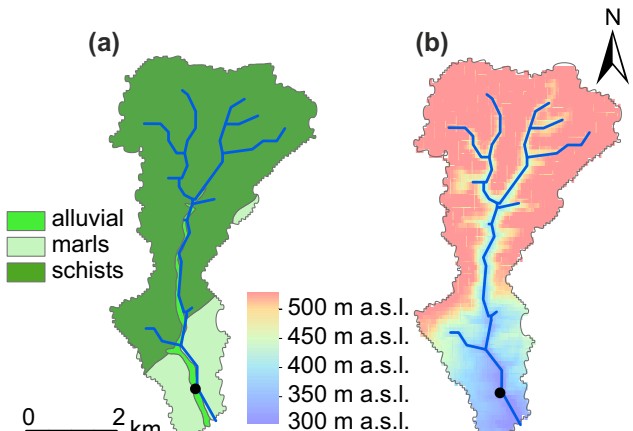

**Figure 1.** Geology and digital elevation model of the Colpach catchment. (a), Geology. (b), Digital elevation model. The black dot marks the sampling location located at the outlet of the catchment (i.e. Colpach-Haut sub-catchment).





## 2.2 Experimental set-up

In order to sample drift diatoms and suspended sediment, the sampling location (Fig. 2) was equipped with two automatic water samplers (ISCO 6712 FS) to collect instantaneous stream water samples (1 L). Sampling was triggered by flow conditions, set up before a rainfall event occurred. During those events, samples were regularly collected each two or three hours. Furthermore, two Phillips samplers were designed as described by Phillips et al. (2000), except that the diameter of the outlet was 2 mm instead of 4. This was done to increase settling of sediment in the sampler, since the catchment average suspended sediment concentrations are relatively small. The samplers were attached close to the river bank and downstream of the automatic samplers. This was done so it would not interfere with the samples of the automatic samplers (Fig. 2). The Phillips samplers were placed on site just before the event and retrieved when the event was completely finished. Together with the placement/removal of the Phillips samplers, we took a manual grab sample (1 L). The Phillips samplers were emptied in 10 L buckets. In addition, turbidity and conductivity were continuously measured in situ at 5 min. intervals using a YSI 600 OMS optical monitoring system, while water depth was measured with an ISCO 4120 pressure probe logger at a 15 min. time-step. Discharge at 15 min. intervals was estimated using a rating curve between discharge and water level. Upon arrival in the lab, all samples were stored at 4°C prior to analysis.

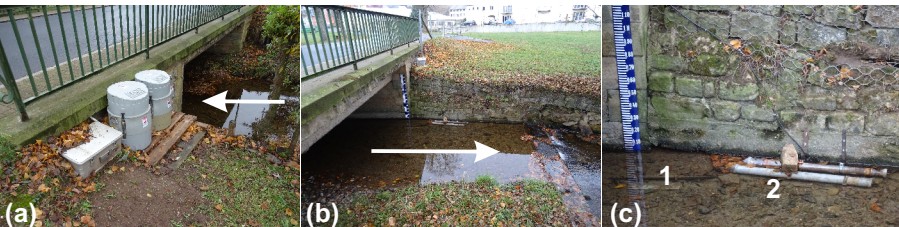

**Figure 2.** Pictures of the experimental set-up at the study location. (a), Automatic samplers installed upstream of the bridge. (b), Overview of the installation of the Phillips samplers. They were installed downstream of the bridge. The arrow denotes stream flow. (c), Close-up of the turbidity meter (1) and the Phillips samplers (2).

### 2.2.1 Diatom preparation and identification

After thoroughly shaking the bottles and buckets, a sub-sample (0.5 L) of each sample was taken for diatom analysis. The sub-samples were fixed with 70 % ethanol to prevent possible population change by cell division and left aside for at least one day to let the diatoms settle down. Diatom slides were prepared for microscopic counts following the CEN 13946 procedure (European Committee for Standardization, 2003). Around 70 % of the supernatant was removed by vacuum aspiration, where after samples were cleaned using $H_2O_2$ and heated (100°C) for 24h in a sand bath. The supernatant was removed by aspiration and 1 mL of HCl (37 %) was added. The mixture was allowed to settle for a day before the supernatant was removed. Afterwards, three repetitions of rinsing with deionized water, decantation and supernatant removal were carried out. The final suspensions were dried on glass cover slips and mounted on permanent slides using Naphrax. A minimum of 400 valves were counted and identified on each slide (n = 103) along random transects using a Leica DMR light microscope with a × 100



oil immersion objective and a magnification of × 1,000. Diatom identifications were mainly based on following taxonomic

135   references: Krammer (2000); Lange-Bertalot (2001); Levkov et al. (2016); Lange-Bertalot et al. (2017).

## 2.3   Suspended sediment analysis

Suspended sediment concentration (SSC) was determined by filtering a known sub-sample volume between 200 and 500 mL

through 1.2 $\mu$m Whatman GF/C glass fibre filters using a Millipore vacuum pump. prior to filtering, the filters were dried

at 105°C and weighted. Afterwards, the samples were dried (105°C) and weighted again. The concentration of suspended

140   sediment is the difference between those two weightings divided by the volume of the filtered sample.

Particle size distribution (PSD) was measured in the laboratory using a portable LISST-200X sensor (Sequoia Scientific, Inc.,

Bellevue, WA), which is a submersible laser-diffraction based particle size analyser. Measurements were done on suspensions

inside a test chamber provided by the manufacturer. Before starting them, a background measurement was done with Milli–Q

Water. After the chamber was filled, the water was stirred using a magnetic stirrer for a few minutes to ensure all bubbles

145   disappeared. If still present, bubbles were removed manually from the measuring cell and windows of the chamber. After

doing the background measurements, the chamber was filled with a 0.5 L sub-sample and Milli–Q water was added if the

sensor was not fully covered. A magnetic stirrer kept all particles in suspension. Then, a LISST measurement consisting of

120 single measurements was performed in real-time modus using the LISST-SOP200X program. The raw data of each single

measurement was then converted to particle size distribution using the 'Random Particle Shape Models' described by Agrawal

et al. (2008) and the recorded background scatter. Unfortunately, it was not possible to measure PSD and SSC in all samples,

due to limited water volumes collected.

## 2.4   Statistical analysis

We performed weighted t-tests with discharge as weight on the particle size (the 10th, 50th and 90th percentiles) and compared

the distributions with a Kolmogorov–Smirnov test. We used discharge as weight, because during periods with higher discharge

a larger volume of water is flown through the Phillips sampler and thus contributes more to the time-integrated samples.

Furthermore, we plotted flow duration curves using the function *fdc* from the R-package *hydroTSM* (Zambrano–Bigiarini,

2017) to check if the events were fully covered by the automated sampling method. We encountered a problem with the

field instrumentation during the third event, as turbidity was exceptionally noisy and measured concentrations too high and

unrealistic for the site (see Fig. 3). Therefore, we decided to exclude the event for further analysis on the sediment data.

Before carrying out the statistical analysis on the diatom communities, we reduced the species dataset by only keeping the

taxa with a relative abundance of at least 1% in minimum two samples. As the species data contains many zeros, we Hellinger

transformed the data and took euclidean distances using the *vegdist* function provided by the R-package *vegan* (Oksanen

et al., 2019). We then performed several hierarchical agglomerative cluster analyses and evaluated them using Gower's (1983)

distances. Eventually, the unweighted arithmetic (UPGMA) cluster analysis showed the best results (i.e. lowest Gower dis-

tance). We used the fusion level values (i.e. dissimilarity values) of the obtained dendrogram to identify the "optimal" number

of clusters. Furthermore, analysis of similarity (ANOSIM) was done on the same transformed dataset to test if communi-





ties significantly differed between the events and sampling methods (Clarke, 1993). This was assessed with the Monte Carlo permutation test (perm-9999).

Next, we calculated diatom biovolume according to the data provided in Rimet and Bouchez (2012) and Omnidia (Lecointe

et al., 1993) as a measure of mean cell size. Also, Shannon–Wiener, Pielou's evenness and specific pollution-sensitivity (IPS; Cemagref (1982)) indices were computed and compared for each method and event using unpaired weighted t-test or Mann–Whitney U test (package *sjstats*) with discharge as weight. The IPS has been chosen out of many diatom-based indices, because this index takes into account the abundance of each species in the sample, their sensitivity value to organic pollution and their indicator value (i.e., relative probability of each species to occur in one of five saprobity classes (i.e. measure of

organic matter present)). In addition, Ecological guilds were assigned to the diatom species following Passy (2007) and Rimet and Bouchez (2012). Ecological guilds are defined as a group of species which live in the same environment (i.e. tolerant to similar environmental stressors), but may have adapted differently (Rimet and Bouchez, 2012). All aforementioned statistical analyses were performed using the R statistical program (R v. 3.5.0.; http://www.r-project.org/).

## 3 Results

The first event occurred after an exceptional dry and warm summer when water levels were very low. Consequently, a maximum catchment discharge of only 0.011 mm h$^{-1}$ was measured (Fig. 3). Turbidity levels also followed the sudden increase in discharge reaching 68.5 NTU. Unfortunately, both the automatic and Phillips samplers were not active during the first peak of the event and this could have influenced our results. The second event happened a few days later and here the discharge, turbidity and SSC increased substantially. A similar pattern in discharge levels as during the first event was seen (i.e. fast

responses to precipitation). This is in strong contrast to the third event where the small peak in discharge is now followed by a high, extended peak (i.e. delayed peak). During this event, we measured the highest catchment discharge (0.44 mm h$^{-1}$) and turbidity levels. However, the latter shows too much noise and therefore we excluded this event for the statistical analysis on the sediment data.





**Figure 3.** Catchment discharge, suspended sediment concentration (SSC), median particle size ($D_{50}$), turbidity and precipitation for the three events. Discharge and turbidity were measured at 15 min. intervals, while SSC and $D_{50}$ were analysed from automatic collected water samples. Total precipitation is given per event. Grey background gives the period the Phillips sampler was in the water.



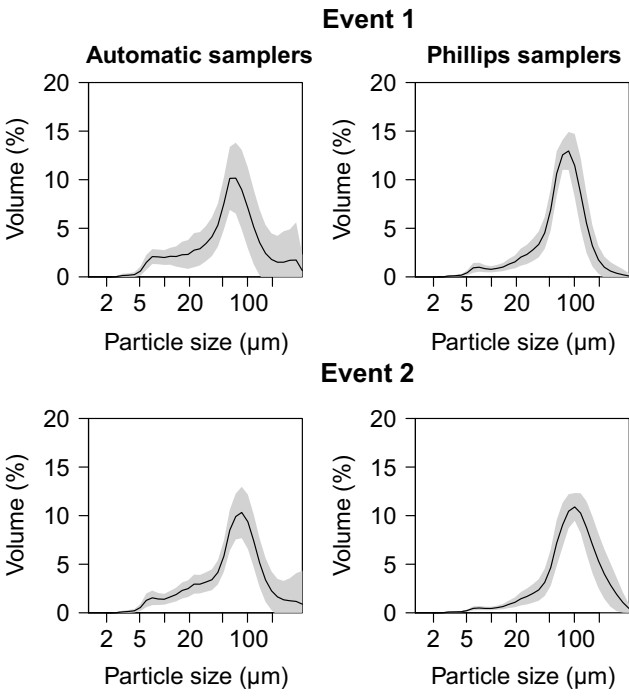

**Figure 4.** Average particle size distribution for the two sampling methods for the first two events. Grey area indicates the standard variation.

**Table 1.** Comparison of the particle size (the 10th, 50th and 90th percentiles) between the Phillips sampler and the automatic sampler. Weighted averages and standard deviations are given. *P*-values were derived from weighted t-tests or Mann–Whitney U tests.

|  | $D_{10}$ ($\mu$m) | $D_{50}$ ($\mu$m) | $D_{90}$ ($\mu$m) |
|---|---|---|---|
| **Event 1** | | | |
| Automatic (n=14) | $61.0 \pm 23.2$ | $82.0 \pm 39.1$ | $124.6 \pm 71.5$ |
| Phillips trap (n=2) | $62.9 \pm 19.1$ | $68.6 \pm 21.4$ | $78.6 \pm 20.4$ |
| *P* value | 0.648 | 0.921 | 0.607 |
| **Event 2** | | | |
| Automatic (n=16) | $53.9 \pm 10.6$ | $62.2 \pm 13.1$ | $79.5 \pm 17.8$ |
| Phillips trap (n=2) | $85.4 \pm 25.1$ | $94.3 \pm 27.5$ | $108.0 \pm 34.8$ |
| *P* value | 0.0297 | 0.0687 | 0.175 |

## 3.1 Suspended sediment

In Figures 3 and 4, median particle sizes ($D_{50}$) and particle size distributions (PSD) are given for the first two events. In general, median particle sizes follow the changes in turbidity levels. However, this is not the case for the last five samples taken during the first storm run-off event. They do not seem correct as also the SSC shows a different behaviour. We do not know yet what these high levels caused, but this effect is also visible in the PSD, where we observe a high standard variation after 200 $\mu$m.





However, we do not observe this second peak in the PSD of the Phillips samples. This difference in the distribution between the

two methods was confirmed by the Kolmogorov–smirnov test (D = 0.31, P = 0.036), while there was no significant difference

for the second event (D = 0.25, P = 0.21). In addition, we observe that particle sizes of the Phillips samples are similar to the

automatic samples (P > 0.05), except for the 10th percentile of the second event (t = -2.39, P = 0.03) (Table 1). Besides, we

notice that particle sizes of the time-integrated method substantially increased for the second event, whereas the mean sizes

decreased for the automated sampling method. Although, the Phillips sampler has a tendency to undersample smaller particles,

it integrates the sediment particle sizes and distributions well on both occasions.

## 3.2 Diatom communities

Generally, the flow duration curves show that the three events were well covered with the automatic samplers, because the

discharge at the time the automatic sampler collected the samples used for diatom analysis (blue line) follows a similar pattern

than the discharge estimated at 15 min. intervals (black line) (Fig. 5). Also, the samples are well distributed over the different

sampling periods. Both results show us that our sampling was not biased towards certain periods of the events.

We identified a total of 233 different taxa, including varieties, subspecies and forms, belonging to 65 genera. After using

the cut-off criteria, 71 species (94 % of the total valves counted) were kept in the statistical analyses. Most species rich genera

were then *Nitzschia*, *Navicula*, *Fragilaria* and *Planothidium* comprising respectively 15, 10, 5 and 4 taxa, whereas *Navicula*

*gregaria* (15 % of the total valves counted), *Navicula germainii* (9 %), *Nitzschia palea* (7.2 %), *Nitzschia tenuis* (6.5 %)

and *Planothidium lanceolatum* (4.2 %) were the most abundant species. An average species richness/sample of 43 ± 6.8 was

observed with 37.9 ± 6.3, 43.7 ± 4.1 and 49 ± 3.9 during the first, second and third event respectively. Terrestrial diatoms were

consistently found to react to the precipitation pulses with the average proportion of terrestrial diatoms in the water samples

increasing to maximum of 6.6 % (event 1), 8.4 % (event 2) and 9.7 % (event 3) during periods of high discharge (see Fig. A1

in Appendix).

Based on the cluster analysis and its respective fusion level values, five different groups could be established (Fig. 6).

Samples of event 3 and 2 are each divided over two groups, whereas almost all samples of event 1 belong to one group. There

is a clear separation between the diatom communities taken during the third event from the ones of the other two events, since

the branches of the dendrogram are very long and splitting occurred early in the process. Concerning the samples taken with

the Phillips sampler, we observe that the ones of the third event are very different than most of the other samples of that event

(both located in a different cluster). Likewise, the time-integrated samples of the second event are quite distinct from each

other, since the samples are also assigned to a different cluster. On the contrary, Phillips samples of event 1 are not really

distinguishable from each other or from the automatic samples. Overall, the cluster analysis groups the diatom communities

relatively well per event with only a few mismatches occurring during the process.

Overall, analysis of similarity (ANOSIM), species abundances (Table 2) and their derived indices (Fig. 7) all confirm the

outcome of the cluster analysis. The ANOSIM enabled the detection and split of the communities based on the different events

(Global R = 0.6433, P = 0.0001). Moreover, there was also a significant difference between the community composition of the

automatic sampler and the Phillips sampler for the third event (R = 0.804, P = 0.002), whereas the communities taken during the





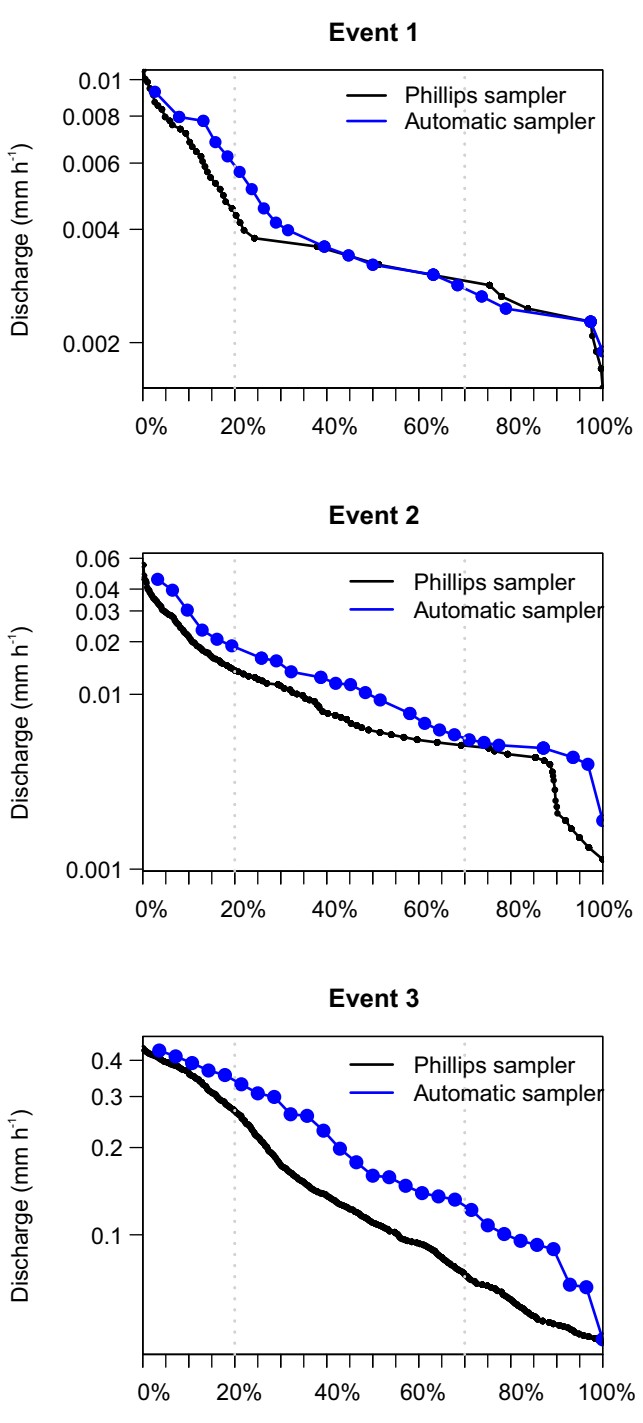

**Figure 5.** Flow duration curves covering the sampling period of the Phillips samplers. Blue line, samples taken with an automatic sampler for diatom analysis. Black line, continuous measurements of discharge at 15 min. intervals.





**Table 2.** Most abundant diatom species for the whole study campaign. The weighted averages (in percent) and standard error (between brackets) are given. Species marked with an "*" indicates that the species relative abundance is significantly different between the two sampling methods for that event.

| Species | Event 1 | | Event 2 | | Event 3 | |
|---|---|---|---|---|---|---|
| | Automatic | Phillips | Automatic | Phillips | Automatic | Phillips |
| | (n = 38) | (n=2) | (n = 31) | (n = 2) | (n = 28) | (n = 2) |
| Navicula germainii[3] | 14.60 ± 0.77 | 11.42 ± 6.24 | 9.28 ± 0.61 | 9.41 ± 0.92 | 1.21 ± 0.18 | 0.00 ± 0.00 |
| Navicula gregaria[3] | 13.89 ± 0.54 | 13.66 ± 0.82 | 18.71 ± 0.51 | 16.01 ± 0.03 | 14.05 ± 0.60 | 8.21 ± 0.10 |
| Nitzschia tenuis[3] | 10.87 ± 0.30 | 8.13 ± 0.06 | 3.54 ± 0.15 | 5.50 ± 0.27 | 0.22 ± 0.07 | 0.00 ± 0.00 |
| Nitzschia palea[3] | 8.99 ± 0.35 | 5.71 ± 0.14 | 9.32 ± 0.15 | 4.04 ± 0.17 | 2.90 ± 0.04 | 1.47 ± 0.00 |
| Navicula cryptocephala[3] | 4.50 ± 0.29 | 2.96 ± 1.74 | 6.09 ± 0.39 | 3.79 ± 0.42 | 2.50 ± 0.18 | 1.72 ± 0.35 |
| Nitzschia dissipata[3] | 3.54 ± 0.38 | 5.77 ± 1.08 | 4.91 ± 0.51 | 4.15 ± 0.50 | 1.35 ± 0.22 | 0.37 ± 0.09 |
| Nitzschia recta[3] | 3.53 ± 0.02 | 1.69 ± 0.00 | 1.12 ± 0.04 | 1.22 ± 0.09 | 0.22 ± 0.09 | 0.00 ± 0.09 |
| Nitzschia linearis[3] | 3.36 ± 0.54 | 1.33 ± 0.38 | 1.81 ± 0.60 | 2.57 ± 1.31 | 0.29 ± 0.28 | 0.00 ± 0.17 |
| Cyclotella meneghiniana[4] | 3.03 ± 0.46* | 10.92 ± 2.59* | 1.76 ± 0.22* | 3.54 ± 0.42* | 0.28 ± 0.05 | 0.12 ± 0.09 |
| Nitzschia acicularis[4] | 2.37 ± 0.29 | 0.25 ± 0.18 | 2.08 ± 0.33 | 0.49 ± 0.35 | 0.12 ± 0.04 | 0.12 ± 0.09 |
| Planothidium lanceolatum[2] | 2.09 ± 0.08 | 2.32 ± 0.00 | 3.02 ± 0.12 | 2.08 ± 0.00 | 9.11 ± 0.13 | 9.20 ± 0.35 |
| Melosira varians[1] | 1.79 ± 0.22 | 2.04 ± 0.38 | 1.86 ± 0.31 | 3.68 ± 1.40 | 1.23 ± 0.25 | 0.00 ± 0.00 |
| Achnanthidium minutissimum[2] | 1.59 ± 0.25 | 3.54 ± 0.68 | 2.59 ± 0.29 | 1.71 ± 0.01 | 8.46 ± 0.46* | 16.16 ± 3.44* |
| Achnanthidium rivulare[2] | 1.46 ± 0.20 | 1.44 ± 0.31 | 2.45 ± 0.26 | 3.79 ± 0.07 | 2.75 ± 0.29 | 7.35 ± 0.68 |
| Reimeria sinuata[2] | 1.41 ± 0.05 | 0.85 ± 0.08 | 1.95 ± 0.17 | 2.81 ± 0.60 | 5.19 ± 0.20* | 1.96 ± 0.62* |
| Cocconeis euglypta[2] | 1.28 ± 0.16 | 1.57 ± 0.22 | 1.26 ± 0.13 | 2.44 ± 0.34 | 3.32 ± 0.24* | 1.10 ± 0.26* |
| Gomphonema parvulum[1] | 1.04 ± 0.13 | 2.21 ± 0.73 | 1.03 ± 0.11 | 1.46 ± 0.51 | 2.84 ± 0.30 | 1.47 ± 0.52 |
| Fragilaria gracilis[4] | 0.88 ± 0.11 | 0.00 ± 0.00 | 2.88 ± 0.18 | 2.94 ± 0.87 | 1.52 ± 0.15 | 0.86 ± 0.26 |
| Planothidium frequentissimum[2] | 0.66 ± 0.24 | 0.72 ± 0.31 | 1.01 ± 0.29 | 1.10 ± 0.43 | 2.51 ± 0.39 | 3.07 ± 1.66 |
| Fragilaria vaucheriae[1] | 0.33 ± 0.10* | 1.57 ± 0.22* | 0.23 ± 0.07 | 0.00 ± 0.00 | 3.35 ± 0.20* | 1.72 ± 0.00* |
| Sellaphora atomoides[3] | 0.19 ± 0.06 | 0.12 ± 0.00 | 1.36 ± 0.06 | 0.85 ± 0.09 | 1.42 ± 0.14* | 5.76 ± 0.60* |
| Sellaphora subseminulum[3] | 0.16 ± 0.35 | 0.00 ± 0.00 | 0.23 ± 0.36 | 0.12 ± 0.12 | 1.54 ± 0.73* | 5.76 ± 0.84* |
| Achnanthidium catenatum[2] | 0.13 ± 0.05 | 0.00 ± 0.00 | 0.51 ± 0.11 | 0.98 ± 0.34 | 0.81 ± 0.14* | 4.41 ± 0.17* |
| Rossithidium petersenii[2] | 0.08 ± 0.21 | 0.00 ± 0.10 | 0.20 ± 0.21 | 0.00 ± 0.08 | 1.16 ± 0.31* | 2.21 ± 0.35* |
| Planothidium daonense[2] | 0.07 ± 0.09 | 0.00 ± 0.16 | 0.06 ± 0.07 | 0.12 ± 0.08 | 0.74 ± 0.16* | 2.08 ± 0.44* |

Ecological guilds: [1] High profile. [2] Low profile. [3] Motile. [4] Planktonic



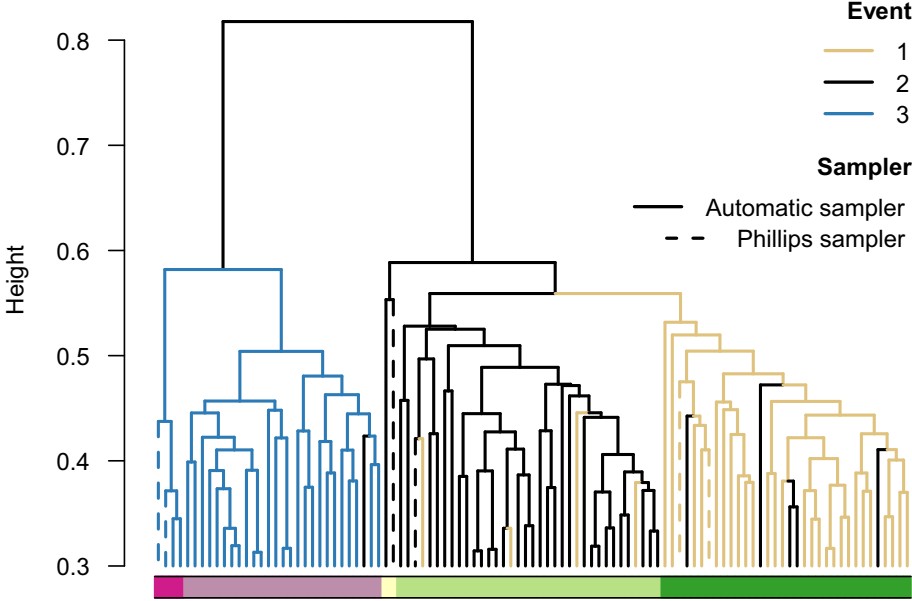

**Figure 6.** Dendrogram of the diatom community data retrieved from unweighted arithmetic (UPGMA) cluster analysis. The y-axis gives the value of the distance metric (i.e. euclidean) between clusters and indicates the similarity between the two groups. The five different clusters are given in the bar below.

other two events could not be separated from each other (event 1: R = 0.361, P = 0.07; event 2: R = 0.129, P = 0.249). A similar outcome is seen when we compare the abundances of the most abundant diatom species with each other. Aside from *Navicula*
*gregaria* and *Navicula cryptocephala* which are dominant throughout the entire study, the abundance of the other dominant taxa differ substantially between the first two and the third event. For instance, *Navicula germainii*, *Nitzschia tenuis* and *Cyclotella meneghiniana* only sparingly occur in the third event, while we observe the reverse for *Planothidium lanceolatum*, *Sellaphora subseminulum* and *Achnanthidium minutissimum*. Interestingly, this follows a shift in ecological guilds going from an assemblage containing more motile species (i.e. unattached free living species immersed on the sediment matrix surrounded
by exopolysaccharides) to more colonial and strongly attached taxa. Regarding the sampling methods, dominant taxa such as *N. germainii*, *N. gregaria*, *N. tenuis* and *Nitzschia palea* vary little in the first two events, except for *C. meneghiniana* and to a lesser extent *Fragilaria vaucheriae* which both occur significantly more in the samples of the Phillips sampler. In contrast to those events, the abundances of *A. minutissimum*, *Reimeria sinuata*, *Sellaphora atomoides*, *F. vaucheriae* and *S. subseminulum* are all very different between the methods for the last event. Of those two communities, the one collected with the automatic
samplers resembles more the communities of event 1 and 2. Concerning the indices, Shannon–Wiener, species evenness and IPS do not show any significant difference between the methods for the first two events (P > 0.05), while only a higher diversity in the automatic samplers is noted for the third event (t = 2.76, df = 28, P = 0.01). Also, the overall diatom size was higher in the automatic samples of the third event compared to the time-integrated samples (Chisq = 7.76, df = 28, P < 0.0001 ), whereas





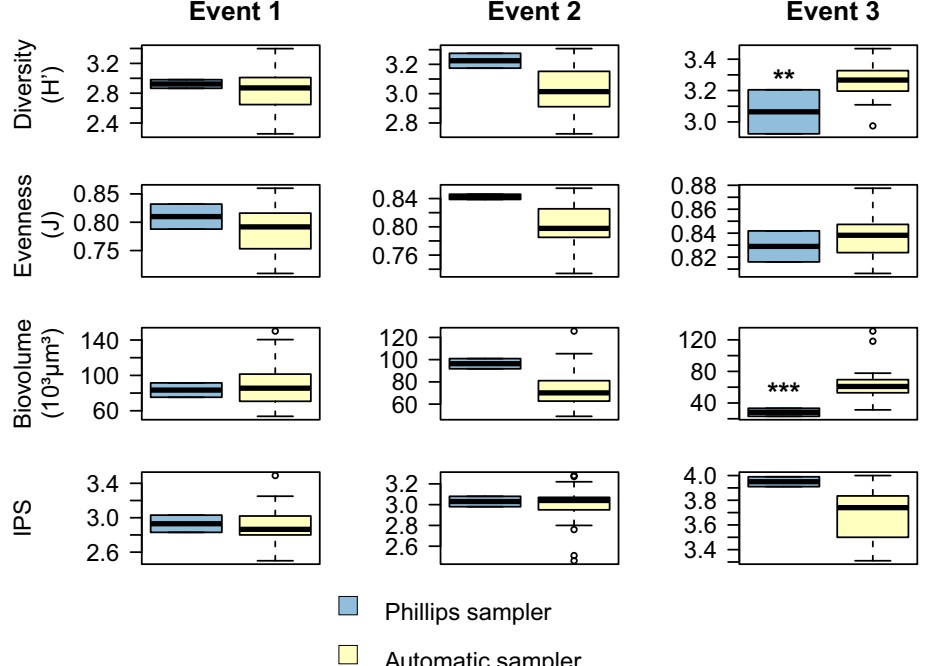

**Figure 7.** Comparison of the diatom community data between the two sampling methods for each event. Shannon–Wiener index (H')
generally ranges between 1.5 (low diversity) and 3.5 (high diversity). Pielou's evenness (J) ranges between 0 (low evenness) and 1 (high
evenness). The specific pollution-sensitivity index (IPS) ranges between 1 (poor water quality) and 5 (high quality). **, $P < 0.01$; ***, $P < 0.001$.

it was not significantly different for the first two events. The results of our analyses indicate that the Phillips sampler could be

a valid tool to collect a time-integrated diatom community representative for the entire sampling period.

## 4 Discussion

### 4.1 Evaluation of the events and samplings

The purpose of this study was to test if the Phillips sampler could be used to collect a time-integrated diatom assemblage as an
alternative to automated sampling methods. To evaluate the sampler, three storm run-off events were simultaneously sampled

with automatic samplers and analysed. The events were well and thoroughly covered as shown in the flow duration curves. Of
those three, two different types of hydrographs were generated and cluster analysis showed that the events could be separated
based on the diatom species composition. Even though we only sampled at one location during a relatively short period (30
October–12 December), we were able to observe quite some variation among events and communities.

Furthermore, previous results of Klaus et al. (2015) and Pfister et al. (2017) confirm that we did sample representative

communities of the stream. Like us, they did an event-based sampling in the Colpach catchment during the same period of year





(both in December). The latter is also important, since diatoms exhibit seasonal succession and thus species composition could change significantly during the year (Wetzel, 2001; Wu et al., 2016). Klaus et al. (2015) analysed 28 samples in which they found 221 different species, while we also observed 231 taxa. Likewise, during the first precipitation pulse the percentage of terrestrial diatoms in the samples increased to 8.9 % at a discharge of around 0.18 mm h$^{-1}$, which is also the proportion that we

found. Of the 15 most abundant species in their study, 12 were also present in our study of which eight were abundant including *N. gregaria*, *P. lanceolatum*, *N. cryptocephala* and *N. linearis*. Similarly, of the 15 dominating species in Pfister et al. (2017), 11 were present in this study of which six were dominant (e.g. *N. gregaria*, *N. palea*, *A. minutissimum*, *P. lanceolatum*). Our research also confirms the observation of Martínez-Carreras et al. (2015), who found that the relative abundance of terrestrial diatoms increased with higher discharge. Unfortunately, since we encountered too much noise in the turbidity and sediment

analysis of the third event, is it difficult to draw robust conclusions from that event. The reasons for this noise are still unknown, and may be linked to a local accumulation of sediment and fine material around and on the sampling tubing of the automatic samplers and a miss-cleaning of the turbidity sensor. But aside from that, results indicate that overall sampling was done properly, that there was some variation between and during the events and that samples were representative for the site.

### 4.2   On the potential for the Phillips sampler to collect drift diatoms

Several statistical analyses were executed on sediment and diatom data to verify if the Phillips sampler could be a useful tool in regards to sampling drift diatoms. As shown here and previously by several other studies, the Phillips sampler gave reasonably good results concerning particle sizes and particle size distributions compared to automated sampling methods (Phillips et al., 2000; Walling, 2005; Perks et al., 2014; Smith and Owens, 2014). Oversampling of bigger sediment particles, as often mentioned in those studies, did occur, but this effect was rather limited in this research. There was also no difference in

the diatom communities and its derivatives such as biovolume and diversity between the two sampling methods for the first two events. However, time-integrated communities sampled during the third event deviated from the assemblages collected with the automatic sampler. This is probably because of the same reason which caused so much noise on the sediment data. Since those samples were collected by means of automated sampling, one could argue that the Phillips sampler should give the better results of the two methods. This is, however, not reflected in the diatom communities, because they actually differed more

from the communities sampled in the previous events. On the other hand, the stream did have a different behaviour, higher discharge and water level compared to the other events and this likely led to the activation, detachment and transportation of species with another behaviour or belonging to a different ecological guild (e.g. more strongly attached species) (see examples in Round et al. (1990); Jewson et al. (2006); Rimet and Bouchez (2012)). Indeed, high motile and planktonic species such as *Navicula* sp. and *Nitzschia* sp. were replaced by colonial and/or attached species (e.g. *Achanthidium* sp. and *Planothidium* sp.)

and this shift was more pronounced in the Phillips sampler. Thus, the Phillips sampler is able to sample a representative diatom community of a storm run-off event, though we were not able to check whether underestimation of certain species occurred during the third event.





### 4.3 Potential way forward

Today, different research avenues can be followed in the exploration of drift diatom sampling methods for hydrological tracing.
Here, the Phillips sampler was successfully applied, though our work was restricted to a single location. Therefore, future studies should test the sampler also in other settings that have a different hydrological behaviour and diatom communities. It would be interesting to investigate if studies would then be able to confirm our results and especially give information on its efficiency at higher discharges, which is still not completely clear from our study. Furthermore, future research should not only be limited to the Phillips sampler. Also, other passive samplers such as the pumped active suspended sediment (PASS) sampler
designed by Doriean et al. (2019) and the bidirectional time-integrated mass-flux sampler (TIMS) developed by Elliott et al. (2017) should be able to collect a representative diatom community. The operating principle of both samplers is the same as the Phillips sampler with the difference that the PASS works at a constant, pre-defined flow rate enabling the measurements of time-weighted average SSC and PSD, while the Bidirectional TIMS is developed for estuaries and has a L-shaped outlet preventing inflow from the other direction. Another possible avenue would be to investigate what the minimum number of
point samples is that we need and at which periods we need to take them (e.g. one at peak discharge, during rising and falling limb) to get a community representing the entire event. Importantly, the diurnal circadian rhythm of benthic diatoms should be taken into account (Müller–Haeckel, 1973; Müller-Haeckel and Håkansson, 1978). Eventually, this information could then significantly reduce the number of samples to be analysed. Besides, molecular techniques (i.e. DNA metabarcoding and High-Throughput Sequencing) in diatom research are developing fast and since these techniques are faster and less expensive than
microscopical examinations, they could also open the path to higher frequency sampling (Vasselon et al., 2017; Rivera et al., 2020). So, in general, this research has unlocked many new possibilities for collecting drift diatoms which could pave the way for better using diatoms in hydrological tracing.

Increased amounts of suspended sediments due to anthropogenic factors can have significant negative impacts on water quality and aquatic biota. Besides that it reduces primary production through a reduction in light penetration, it also acts as an
important vector for the transfer of nutrients and metals in fluvial systems (Bowes et al., 2003; Ballantine et al., 2008; Bilotta and Brazier, 2008). Therefore, it is important to identify the sediment sources, what their relative contribution is to the overall sediment load and to know the effect on the ecological functioning of the system so that remedial measures could be taken. A way to investigate this is with sediment fingerprinting. There, several tracers (e.g. mineralogy, nitrogen and carbon stable isotopes) are used to serve as sediment fingerprints (Haddadchi et al., 2013). As (drift) diatoms are often associated with those
particles, they could provide additional information on sediment quality, its sources and transport (Jewson et al., 2006; Jones et al., 2012). More interestingly, they could even be sampled and analysed together with the suspended sediment to determine potential (sediment) sources and their relative contribution in a catchment (Pfister et al., 2017). Besides, the use of a time-integrated sampler enables us also to collect a phytoplankton community, even at different depths (McDonald et al., 2010). Generally, in water quality assessment, phytoplankton is sampled with grab samples (Soylu and Gönülol, 2003; Abonyi et al.,
2012). However, the composition of planktonic diatoms is quite variable in time and changes occur relatively fast in comparison to benthic communities (Round et al., 1990). Therefore, it would be interesting to test if a time-integrated suspended sediment

sampler would also give a representative phytoplankton community for a certain period of time. So, the use of a time-integrated mass-flux sampler could not only assist sediment fingerprinting, but potentially also improve the analysis of water quality.

## 5 Conclusions

Here, we investigated if the Phillips sampler, a time-integrated mass-flux sampler, was able to sample representative diatom communities during a storm run-off event. This was done by comparing the suspended sediment concentrations, particle size distributions and drift diatom assemblages with point samples collected by automatic samplers. Our results indicate that during two storm run-off events the Phillips sampler sampled representative samples, whereas significantly different communities were collected during the third event. However, sediment data of this event, which was sampled with automatic samplers, showed
much noise meaning that we could not verify if the Phillips sampler sampled representative communities or not. Nevertheless, we believe that this sampler could not only be applied in hydrological tracing using terrestrial diatoms, but may also be a useful tool in water quality assessment.

*Data availability.* Foets, J., Martinez-Carreras, N., Iffly, J., Pfister, L.: A time-integrated sediment trap to sample diatoms for hydrological tracing, Mendeley Data, V1, doi: 10.17632/hgbf5bpwkh.1 https://data.mendeley.com/datasets/hgbf5bpwkh/draft?a=961d56b1-f410-4201-
b0fe-c1d80dca84c4, 2020

*Author contributions.* JF, CEW, AJT, NMC and LP designed and directed the study. JFI planned and carried out the field work. NMC Carried out the analysis of the sediment data. JF prepared the diatom slides and carried out diatom identifications. JF, CEW, NMC, LP, and AJT discussed and interpreted the results. JF prepared the paper with contributions from all co-authors

*Competing interests.* The authors declare that they have no conflict of interest.

*Financial support.* This work was supported by the Luxembourg National Research Fund (FNR) (PRIDE15/10623093/HYDRO-CSI).

## Appendix A



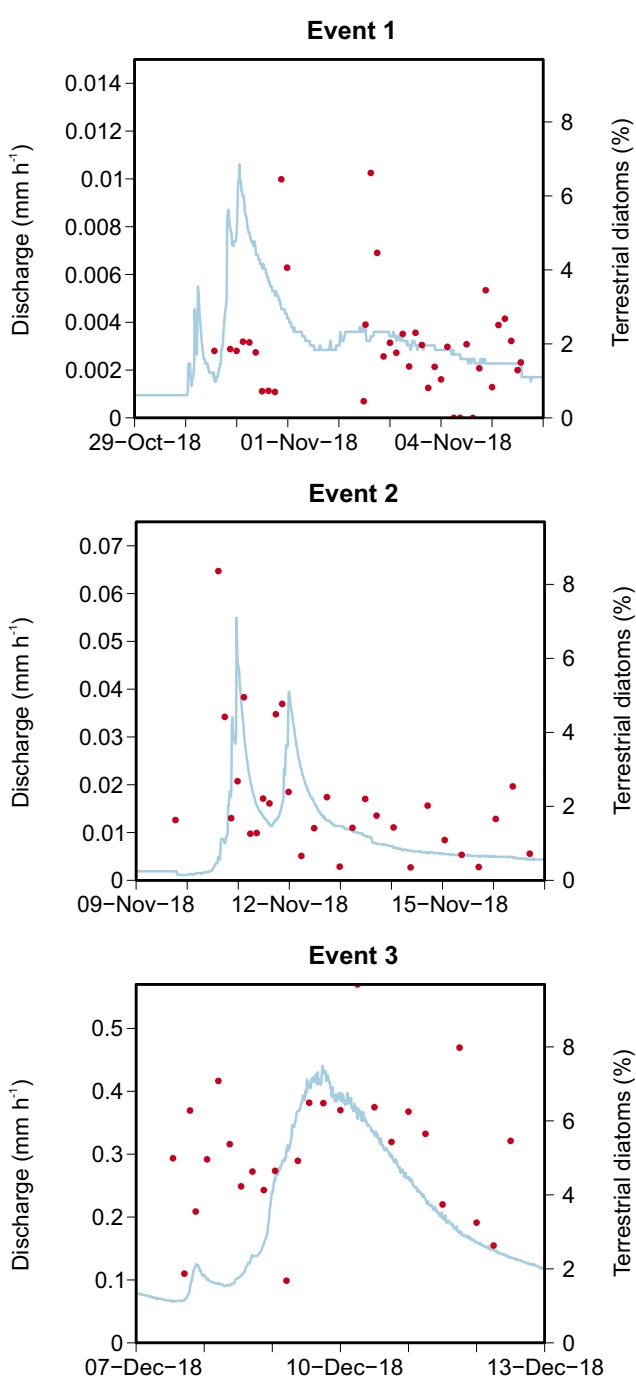

**Figure A1.** Discharge and percentage of terrestrial diatoms in the community for the three storm run-off events.



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
