# Peer review of "Technical note: A time-integrated sediment trap to sample diatoms for hydrological tracing"

_Hydrology and Earth System Sciences, 2020_

## Referee Comment (RC1) · Anonymous Referee #1 · 6 Jun 2020

General comments: In general, the manuscript is well-written, well-structured, and well-referenced. The language is fluent and easy to follow. The authors present a novel concept and tool for hydrological tracing examination. I believe it will be attractive not only by the hydrological community but also by broader readers. Specific comments: Abstract: Half of the abstract are talking about the background and the importance of the study. Limited contents reflect the methods and results of this study. Results: Regarding the goal of treating diatoms as a tracer, I assume that the terrestrial species in the samples are especially important for evaluating the events and the capture rates of the method. I would recommend offering a list of the species list in the Appendix. In addition, regardless of the characteristics of the dominant species in the samples, changes in the terrestrial diatoms would be meaningful as well. I realize that the au-

thors provide a figure for the percentage of the terrestrial diatom, while without telling which kind of percentage it is. Is that the species richness changes or abundance or biomass? Whether they have same or different percentages from the two sampling methods? Discussion: I really like the introduction part which fully links the sediments and diatoms, and the reason why the diatoms species-specific traits could influence the sampling results. However, its pity that there are no further relevant answers to them either in results or discussion part. I would suggest that it could be extended and supplemented regarding the findings of the present study. Technical corrections (typing errors): Line 336, "JFI" may change to "JF".

---

## Referee Comment (RC2) · Francis Burdon (Referee) · 23 Jun 2020

Studying fluvial processes and sediment dynamics is important to understanding key sources of environmental stress in stream and rivers. This study advances our methodological approaches to sampling suspended sediment, characterizing flood attributes, and source-tracking inputs of sediment using diatom assemblages. The latter point is vitally important in understanding when, where, and how fine sediment is mobilized and exported in stream-riparian networks. The Authors compared suspended sediment sampling methods using Phillips and ISCO automated samplers over the course of three high-flow events in the same stream. A novel aspect builds on previous research by investigating the potential for terrestrial and aquatic diatom assemblages in the captured samples to indicate event severity and trace the sources of transported

sediment. The study is well executed and clearly written, with the discussion highlighting the state-of-the-art in sediment research and how complementary methods to the present research could be used to better quantify sediment inputs across the land-water interface. Overall, I think this study is useful and should definitely be reported as a Technical Note demonstrating the results as a proof of concept.

However, I do have some concerns about the data analysis. Over the three sampling events there is a great mismatch in the number of samples used for each sampling method. This concerns me for two reasons. Firstly, low sample replication for the method using the Phillips sampler increases the chance of a Type II error (i.e., not detecting a difference where there is one). Secondly, the analyses involve comparing samples collected over the entire duration of the event (Phillips sampler) with samples collected in three-hourly intervals (Automatic sampler). This is problematic for two reasons – the samples are not directly equivalent, and the automatic samples are non-independent (i.e., temporally auto-correlated). Below are some more points relating to these aspects and other comments I have.

1. In each sampling event, the Phillips samples seem to be at the periphery of each cluster (Fig.6), suggesting that there is some systematic bias in assemblages collected, yet the ANOSIM results suggest that in the first two events, samples were representative (i.e., not significantly different). With only two samples to compare with 38 automatic samples per sampling event it is likely that low statistical power increases the chance of a Type II error (although by not pooling the automatic samples actually inflates the replication – akin to pseudoreplication). I think the authors need to be more cautious with the inferences made (e.g., L228, L285, etc.).

2. I thought the automatic samplers might be pooled over time so that the comparison between these and the Phillips samples are equivalent, but they are not. Is it not a problem to compare the automatic samples collected every 3 hours with a couple of time-integrated samples using the different sampling method? The weighting using discharge for time-integrated samples in the Mann-Whitney U tests (L153-155,

L173) probably helps control for differences across events but it is not clear if the same principle is applied for the cumulative discharge and automatic samples (i.e., temporal auto-correlation of samples collected progressively through the flow event).

3. L160 - Removing rare and uncommon taxa is contentious for multivariate community analysis. See Cao et al. (2001) for more regarding the potential issues. Perhaps the authors can better explain why they did this and what (if any) influence it had on their results. Cao, Y., Larsen, D., & Thorne, R. (2001). Rare species in multivariate analysis for bioassessment: Some considerations. Journal of the North American Benthological Society, 20, 144-153. doi:10.2307/1468195

4. L163-167 - The cluster analysis appears to follow practices recommended by Borcard et al. (2011), but Quinn and Keough (2002) highlight one disadvantage of agglomerative cluster analysis relating to the interpretability of the dendogram. Essentially because the hierarchical approach forms clusters that cannot be later broken, the dendogram is not a representation of all pairwise dissimilarities in objects like in multidimensional scaling. Thus, it could be useful to visualize a MDS plot of the data to determine their relative dissimilarity – also opening up the potential to use PERMANOVA ("adonis") to test differences (which has the advantage of being less susceptible to dispersion effects than ANOSIM). That approach has the advantage of using a "strata" term for event and just testing the overall difference between sampling methods. Using a SIMPER analysis could help bolster the observations made at L283-285 about why the sampling methods differ (i.e., there is some systematic bias for certain taxa). If the authors see this as useful, I would also strongly consider pooling the automatic samples (power issues notwithstanding) and using relative abundances. The removal of rare taxa is probably essential here since the much greater effort identifying diatoms for a pooled automatic sample increases the probability of detecting rare taxa. Borcard, D., et al. (2011). Numerical Ecology with R. New York, NY, Springer New York. Quinn, G. and M. Keough (2002). Experimental Design and Data Analysis for Biologists. Cambridge, UK, Cambridge University Press.

---

## Author Comment (AC1) · 6 Jul 2020

Dear Referee, We thank you for your valuable and constructive comments. We agree with most of your suggestions and remarks and we will adjust accordingly. However, if we would pool the data then we will compare just one sample against another. It seems to us that it is better to include all the samples and take into account the repeated measurements by using a 'strata' term when setting the permutations of the analysis. Also, by using different kinds of analysis and giving the variability of the samples (i.e. boxplot and standard deviation) we believe that it should be sufficient. Besides, could you elaborate a bit more on comment number 2? Especially, the last sentence of the paragraph is unclear to us and we do not know what you suggest.

---

## Author Comment (AC2) · 10 Jul 2020

**Author response to RC1 on "Technical note: A time-integrated sediment trap to sample diatoms for hydrological tracing" by Jasper Foets et al.**

**Jasper Foets on behalf of all co-authors**

We are grateful to the referee for the constructive and valuable comments. Below we reply on the suggestions/comments.

We agree with all his/her comments and will put more emphasis on the results in the abstract, add in the appendix a full species list and specify Figure 7 in the appendix.

Specific future changes:

- Add "relative abundances" to Table 2 and Figure 7
- Add sentences in the discussion referring to the differences between diatoms and suspended sediment particles
- Refer to the different behaviour of diatoms compared to suspended sediment in the statistics section linking it better with the use of ecological guilds.
- Modify the abstract substantially, focusing more on the results and less on the introduction.
- JFI refers to Jean-François Iffly, one of the co-authors.

---

## Author Comment (AC3) · 10 Jul 2020

**Author response to RC2 on "Technical note: A time-integrated sediment trap to sample diatoms for hydrological tracing" by Jasper Foets et al.**

**Jasper Foets on behalf of all co-authors**

We are very grateful to the referee for the constructive and valuable comments. Below we reply on the suggestions/comments.

1) We will adjust some sentences in the text, so that we will be more careful with our conclusions. Previous studies already suggested that comparing time-integrated methods with point samples is rather difficult as barely any statistical analyses were involved (Doriean et al., 2019; Martin et al., 2003; Phillips et al., 2000; Smith and Owens, 2014).

2) To counteract for potential temporal auto-correlation, we will run a NMDS analysis (figure will be added in appendix or text) and compare the position of the centroids in the geometric framework of the NMDS plot using PERMANOVA.

3) We removed rare taxa because occurrences of rare species are likely a matter of chance rather than an ecological meaning. Besides, diatom community data is very variable. We recognize that rare species can have valuable information, but here we were looking for the general patterns and therefore we decided to remove them. In addition, removal of rare species does not affect indices such as Shannon-Wiener or Pielou's evenness index according to (Yu et al., 2017).

4) There is a great mismatch in the number of samples collected with the two methods. However, we do not agree with pooling the data, since we then do not have replicates, even though our analysis is susceptible to pseudo-replication. As an alternative approach, we will specify nestedness in the PERMANOVA model. If there would be a significant effect of treatments (i.e. sampling methods), we will test for dispersion as well. In addition, we will do SIMPER tests on the community data. The statistical analysis of the study will now involve NMDS analysis, PERMANOVA, cluster analysis, ANOSIM, SIMPER and comparison of species relative abundances and derived indices using weighted t-tests or Mann-Withney U test. Below, we attached the outcome of the NMDS, which does not seem to deviate much from the dendrogram approach.

In addition to the changes requested by both reviewers, we will change "Foets et al. in review" since it has been published in the meantime.

Doriean, N.J.C., Teasdale, P.R., Welsh, D.T., Brooks, A.P., Bennett, W.W., 2019. Evaluation of a simple, inexpensive, in situ sampler for measuring time-weighted average concentrations of suspended sediment in rivers and streams. Hydrol. Process. 33, 678–686. https://doi.org/10.1002/hyp.13353

Martin, H., Patterson, B.M., Davis, G.B., Grathwohl, P., 2003. Field trial of contaminant groundwater monitoring: Comparing time-integrating ceramic dosimeters and conventional water sampling. Environ. Sci. Technol. 37, 1360–1364. https://doi.org/10.1021/es026067z

Phillips, J.M., Russell, M.A., Walling, D.E., 2000. Time-integrated sampling of fluvial suspended sediment: A simple methodology for small catchments. Hydrol. Process. 14, 2589–2602. https://doi.org/10.1002/1099-1085(20001015)14:14<2589::AID-HYP94>3.0.CO;2-D

Smith, T.B., Owens, P.N., 2014. Flume- and field-based evaluation of a time-integrated suspended sediment sampler for the analysis of sediment properties. Earth Surf. Process. Landforms 39, 1197–1207. https://doi.org/10.1002/esp.3528

Yu, Z., Wang, H., Meng, J., Miao, M., Kong, Q., Wang, R., Liu, J., 2017. Quantifying the responses of

biological indices to rare macroinvertebrate taxa exclusion: Does excluding more rare taxa cause more error? Ecol. Evol. 7, 1583–1591. https://doi.org/10.1002/ece3.2798

---

## Author Response (AR1)

**Author response on "Technical note: A time-integrated sediment trap to sample diatoms for hydrological tracing" by Jasper Foets et al.**

**Jasper Foets on behalf of all co-authors**

We are grateful to both referees for their constructive and valuable comments. Below we reply on the suggestions/comments of the reviewers.

Referee1:

*Abstract: Half of the abstract are talking about the background and the importance of the study. Limited contents reflect the methods and results of this study*

Modified the abstract substantially, focusing more on the results and less on the introduction.

*Results: Regarding the goal of treating diatoms as a tracer, I assume that the terrestrial species in the samples are especially important for evaluating the events and the capture rates of the method. I would recommend offering a list of the species list in the Appendix. In addition, regardless of the characteristics of the dominant species in the samples, changes in the terrestrial diatoms would be meaningful as well*

We added a full species list with species relative abundances per event in the appendix.

*I realize that the auC1 HESSD Interactive comment Printer-friendly version Discussion paper thors provide a figure for the percentage of the terrestrial diatom, while without telling which kind of percentage it is. Is that the species richness changes or abundance or biomass? Whether they have same or different percentages from the two sampling methods?*

Table 2 and Figure 7 shows the diatom relative abundances, so we added this to the captions.

*Discussion: I really like the introduction part which fully links the sediments and diatoms, and the reason why the diatoms species-specific traits could influence the sampling results. However, its pity that there are no further relevant answers to them either in results or discussion part. I would suggest that it could be extended and supplemented regarding the findings of the present study.*

We added a few sentences in the discussion referring to the differences between diatoms and suspended sediment particles. In addition, we referred to the different behaviour of diatoms compared to suspended sediment in the statistics section linking it better with the use of ecological guilds.

*Line 336, "JFI" may change to "JF".*

This refers to one of the co-authors: Jean-François Iffly

Referee2:

*In each sampling event, the Phillips samples seem to be at the periphery of each cluster (Fig.6), suggesting that there is some systematic bias in assemblages collected, yet the ANOSIM results suggest that in the first two events, samples were representative (i.e., not significantly different). With only two samples to compare with 38 automatic samples per sampling event it is likely that low statistical power increases the chance of a Type II error (although by not pooling the automatic samples actually inflates the replication – akin to pseudoreplication). I think the authors need to be more cautious with the inferences made (e.g., L228, L285, etc.).*

We adjusted some sentences in the text, so that we are more careful with our conclusions.

*I thought the automatic samplers might be pooled over time so that the comparison between these and the Phillips samples are equivalent, but they are not. Is it not a problem to compare the automatic samples collected every 3 hours with a couple of time-integrated samples using the different sampling method? The weighting using discharge for time-integrated samples in the Mann-Whitney U tests (L153-155, L173) probably helps control for differences across events but it is not clear if the same principle is applied for the cumulative discharge and automatic samples (i.e., temporal auto-correlation of samples collected progressively through the flow event).*

We did not agree with pooling the samples, since we would lose too much information on the variability of the different samples. To counteract for potential temporal auto-correlation, we ran a NMDS analysis (figure is added in the text) and compared the position of the centroids in the geometric framework of the NMDS plot using PERMANOVA.

*L160 - Removing rare and uncommon taxa is contentious for multivariate community analysis. See Cao et al. (2001) for more regarding the potential issues. Perhaps the authors can better explain why they did this and what (if any) influence it had on their results. Cao, Y., Larsen, D., & Thorne, R. (2001). Rare species in multivariate analysis for bioassessment: Some considerations. Journal of the North American Benthological Society, 20, 144-153. doi:10.2307/1468195*

We kept our analysis excluding rare taxa (see authors response to RC2).

*L163-167 - The cluster analysis appears to follow practices recommended by Borcard et al. (2011), but Quinn and Keough (2002) highlight one disadvantage of agglomerative cluster analysis relating to the interpretability of the dendogram. Essentially because the hierarchical approach forms clusters that cannot be later broken, the dendogram is not a representation of all pairwise dissimilarities in objects like in multidimensional scaling. Thus, it could be useful to visualize a MDS plot of the data to determine their relative dissimilarity – also opening up the potential to use PERMANOVA ("adonis") to test differences (which has the advantage of being less susceptible to dispersion effects than ANOSIM). That approach has the advantage of using a "strata" term for event and just testing the overall difference between sampling methods. Using a SIMPER analysis could help bolster the observations made at L283-285 about why the sampling methods differ (i.e., there is some systematic bias for certain taxa). If the authors see this as useful, I would also strongly consider pooling the automatic samples (power issues notwithstanding) and using relative abundances. The removal of rare taxa is probably essential here since the much greater effort identifying diatoms for a pooled automatic sample increases the probability of detecting rare taxa. Borcard, D., et al. (2011). Numerical Ecology with R. New York, NY, Springer New York. Quinn, G. and M. Keough (2002). Experimental Design and Data Analysis for Biologists. Cambridge, UK, Cambridge University Press.*

We did not pool the data, since we then do not have replicates. As an alternative approach, we specified nestedness in the permanova model. There was a significant effect of treatments (i.e. sampling methods) and thus we tested for dispersion as well. In addition, we did SIMPER tests on the community data. We removed the cluster analysis, since the samples were dependent and temporal auto-correlated making cluster analysis problematic. NMDS and PERMANOVA replaces it. The statistical analysis of the study now involves NMDS, PERMANOVA, ANOSIM, SIMPER and comparison of species relative abundances and derived indices using weighted t-tests or Mann-Withney U test.

[revised manuscript text omitted]